# Propagating wave in a fluid by coherent motion of 2D colloids

Koki Sano 1,2,5, Xiang Wang[1], Zhifang Sun[1], Satoshi Aya 1, Fumito Araoka 1, Yasuo Ebina 3, Takayoshi Sasaki 3, Yasuhiro Ishida 1✉ & Takuzo Aida 1,4✉

Just like in living organisms, if precise coherent operation of tiny movable components is possible, one may generate a macroscopic mechanical motion. Here we report that ~$10^{10}$ pieces of colloidally dispersed nanosheets in aqueous media can be made to operate coherently to generate a propagating macroscopic wave under a non-equilibrium state. The nanosheets are initially forced to adopt a monodomain cofacial geometry with a large and uniform plane-to-plane distance of ~420 nm, where they are strongly correlated by competitive electrostatic repulsion and van der Waals attraction. When the electrostatic repulsion is progressively attenuated by the addition of ionic species, the nanosheets sequentially undergo coherent motions, generating a propagating wave. This elaborate wave in time and space can transport microparticles over a long distance in uniform direction and velocity. The present discovery may provide a general principle for the design of macroscopically movable devices from huge numbers of tiny components.

[1] RIKEN Center for Emergent Matter Science, 2-1 Hirosawa, Wako, Saitama 351-0198, Japan. [2] JST PRESTO, 4-1-8 Honcho, Kawaguchi, Saitama 332-0012, Japan. [3] National Institute for Materials Science, International Center for Materials Nanoarchitectonics, 1-1 Namiki, Tsukuba, Ibaraki 305-0044, Japan. [4] Department of Chemistry and Biotechnology, School of Engineering, The University of Tokyo, 7-3-1 Hongo, Bunkyo-ku, Tokyo 113-8656, Japan. [5] Present address: Department of Chemistry and Materials, Faculty of Textile Science and Technology, Shinshu University, 3-15-1 Tokida, Ueda, Nagano 386-8567, Japan. ✉email: y-ishida@riken.jp; aida@macro.t.u-tokyo.ac.jp

Coherent operation under a non-equilibrium state is essential in living organisms for amplifying tiny molecular-scale motions of their movable components into a macroscopic mechanical force[1–3]. For such coherent operation to occur in living organisms, a large number of movable components are positioned in a spatially specific manner so that they can cooperate with each other for large-scale synchronization. Inspired by biological coherent systems, several pioneering attempts have been made to conjugate multiple synthetic movable units for the creation of a mechanical motion[4–26]. A straightforward approach is to connect them with chemical or topological bonds[4–12], but the number of connectable units is practically limited, so that individual motions cannot be amplified into a macroscopic motion. To overcome this issue, we employed competitive repulsive and attractive forces between the movable components for strongly correlating them over a long range. We succeeded in correlating ~$10^{10}$ individual pieces of colloidal nanosheets dispersed in water to form a uniformly separated (~420 nm) cofacial assembly via their competitive electrostatic repulsion and van der Waals attraction. When the electrostatic force is progressively attenuated by the addition of ionic species, we found that the nanosheets sequentially undergo coherent motions, generating an autonomously propagating wave (Fig. 1 and Supplementary Movie 1). This wave can transport microparticles over a long distance in uniform direction and velocity. The wave formation mechanism can be elucidated by the Helfrich–Hurault instability[27–33], which is known to occur in liquid crystalline systems of mesogenic molecules. The present system makes use of a strategy of remote and direct communication between movable components, which is fundamentally different than most reported examples that are assisted by condensed structured media[13–19].

## Results and discussion

### Enhancement of the correlation between titanate nanosheets (TiNSs) in water

The key substance that enabled our achievement is a unilamellar TiNS[34,35], which is characterized by its ultra-thin (0.75 nm) and extra-wide (~5 μm) dimensions (Supplementary Fig. 1). TiNS carries dense negative charges (1.5 C m$^{-2}$) and therefore can be colloidally stable in aqueous media. As described by the Derjaguin–Landau–Verwey–Overbeek (DLVO) theory[36,37], colloidally dispersed TiNSs are correlated with each other by electrostatic repulsion in competition with van der Waals attraction, so that they adopt a cofacial geometry with a large and uniform distance. As previously reported[37], the interaction between TiNSs can be highly strengthened by thorough deionization of a TiNS dispersion. In a 10 T magnetic field, colloidally dispersed TiNSs ([TiNS] = 0.5 wt%) align perpendicular to the magnetic flux lines, thereby forming a monodomain structure with a long-range geometrical integrity (Fig. 1a, left). This structural feature was confirmed by small-angle X-ray scattering[37–39] and scanning electron microscopy (SEM; Supplementary Fig. 2). Owing to the large and uniform TiNS distance of 410 nm, as estimated by SEM imaging, the TiNS dispersion exhibited a vivid green structural color. It is intriguing that such a single crystal-like structural integrity with size regimes of up to several centimeters is readily formed in fluidic media. This monodomain structure, once generated, is highly stable, owing to the strong correlation between TiNSs and barely relaxes even over 2 days after turning off the magnetic field.

### Generation of a TiNS-based propagating wave in water

We then investigated how TiNSs behave when their single crystal-like monodomain assembly is chemically perturbed. According to the DLVO theory, the addition of ionic species can attenuate the electrostatic repulsion between TiNSs, thereby reducing the distance between TiNSs (Fig. 1b and Supplementary Fig. 3). To progressively attenuate the electrostatic repulsion between TiNSs in their monodomain assembly ([TiNS] = 0.5 wt%; quartz cuvette size = 40 × 10 × 1 mm), we gently introduced an aqueous solution of NaCl from the open end of the cuvette, so that NaCl diffused toward the opposite end of the cuvette. Surprisingly, we observed that a wave emerged at the open end of the cuvette and propagated autonomously towards the opposite end (Supplementary Fig. 4a). In addition to NaCl, other ion sources such as acids, bases and even gaseous $CO_2$ generated the propagating wave (Supplementary Fig. 4). For the purpose of precise characterization of the wave, we employed atmospheric $CO_2$ (0.04% in air) as the source for ions ($HCO_3^-$, $CO_3^{2-}$, and $H_3O^+$), because it could gradually penetrate from a slightly opened screw cap into the cuvette interior without mechanical deterioration of the TiNS orientation. As shown in Fig. 1c, the wave propagated from the open end of the cuvette at a velocity of ~150 μm h$^{-1}$ (Supplementary Movie 1), continued for >20 h, and finally ceased when the wave front reached the opposite end of the cuvette (Supplementary Fig. 5). The wave thus generated showed a full wavelength of ~200 μm and contained stable dislocation defects, although their density was low (<2 mm$^{-2}$). As expected, when the cuvette was placed in a $N_2$-filled container, no wave was detected and the monodomain assembly was maintained for >20 h (Supplementary Movie 2). Meanwhile, in a pure $CO_2$ atmosphere, the wave propagation took place more quickly, but its duration time is ~2 h because of the higher concentration of $CO_2$ (Supplementary Fig. 4e and Supplementary Movie 3).

### Spatiotemporal characterization of the propagating wave

We successfully confirmed the wave structure of TiNSs by fixing their spatial orientation using two-stage in-situ polymerization of an acrylic monomer and a silica source. As shown in Fig. 2a and Supplementary Fig. 6, the SEM images showed that the frozen wave thus obtained had a wavelength of ~200 μm, which is consistent with that obtained by optical microscopy (Fig. 1c). These observations indicate that, in the cross section of the wave along the zx-plane, TiNSs were sinusoidally directed along the x-axis (Fig. 2d). Polarized optical microscopy (POM) under crossed Nicols also supported this wave structure (Fig. 2b). By a POM-based retardation measurement with a Berek compensator, the maximum tilting angle of the TiNS plane ($\theta_{max}$) was determined to be 22°. Note that confocal laser scanning microscopy (CLSM) in a reflection mode can selectively visualize TiNSs that are oriented parallel to the xy-plane, thereby showing the 3D structure of the wave[39]. Figure 2c shows a 1-mm-thick 3D structure of the visualized wave, where regions (i) (z = 0–0.2 mm) and (v) (z = 0.8–1.0 mm) appear to be bright, because TiNSs located in these regions are oriented parallel to the xy-plane due to the surface anchoring effect of the cuvette. Meanwhile, in regions (ii)–(iv) (z = 0.2–0.8 mm), thin bright layers appear periodically along the x-axis, which correspond to the peak and trough regions of the wave (Fig. 2c, right). The spatiotemporal POM profiles (Fig. 3a–c and Supplementary Movie 4) and time-dependent CLSM images (Fig. 3d and Supplementary Movie 5) show that the propagating wave always retains its high structural integrity. It is worth noting that the motions of a huge number (~$2.4 \times 10^{10}$ pieces) of TiNSs are governed to be synchronous by a single rule, both in time and space, to generate a macroscopic motion under the present experimental conditions ([TiNS] = 0.5 wt%; observed volume = ~10 mm × 10 mm × 1 mm).

### Formation mechanism of the propagating wave

This intriguing wave propagation is considered to occur according to the following mechanism. First, when $CO_2$ is allowed to penetrate from the slightly opened screw cap inlet into the interior of the cuvette, ions such as $HCO_3^-$, $CO_3^{2-}$, and $H_3O^+$ are produced in the aqueous TiNS

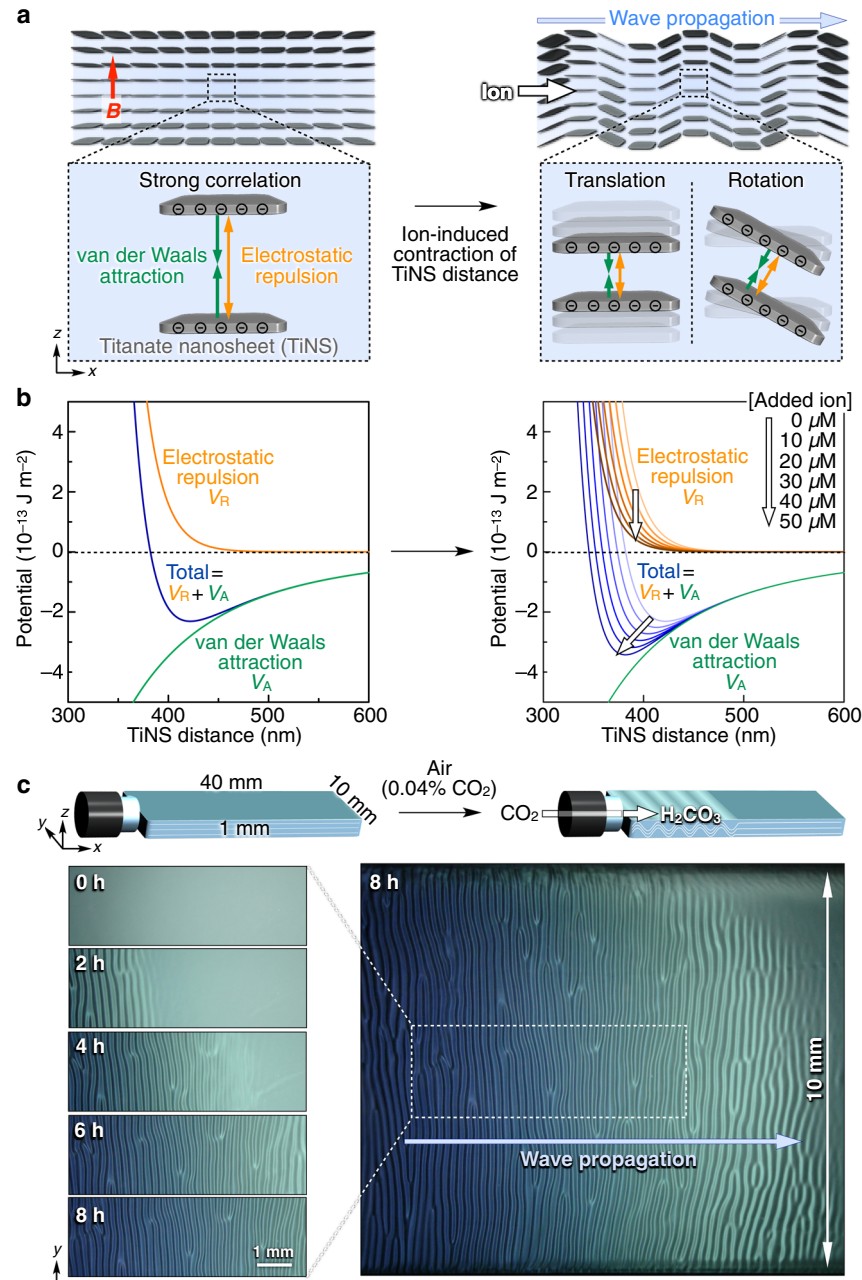

**Fig. 1 Autonomously propagating wave generated by the coherent motion of titanate nanosheets (TiNSs) in their aqueous dispersion. a** Schematic illustrations of wave propagation by the coherent motion of negatively charged titanate nanosheets (TiNSs). In water, colloidally dispersed TiNSs are strongly correlated with each other, adopting a cofacial geometry with competitive electrostatic repulsion and van der Waals attraction. When a 10 T magnetic field is applied to this aqueous dispersion in a quartz cuvette along its $z$-axis, all the nanosheets align parallel to the $xy$-plane of the cuvette and form a monodomain cofacial assembly (left). After the magnetic field is turned off, ionic species are added from the open end of the cuvette, generating an ion gradient that progressively attenuates the electrostatic repulsion between TiNSs. Consequently, the van der Waals attraction between TiNSs prevails, thereby causing a contraction of the TiNS distance through their tilting motion. Owing to the strong correlation between TiNSs, this colloidal motion occurs coherently over a large scale, generating a wave, which then propagates unidirectionally along the direction of ion diffusion (right). **b** Ion-induced contraction of the TiNS distance elucidated by the DLVO theory. Theoretically calculated total potential ($V_A + V_R$) at different ion concentrations (0.27–0.32 mM) is plotted as a function of the TiNS distance. When ionic species are added to the colloidal dispersion, the electrostatic repulsion between TiNSs is attenuated due to the screening effect, thereby reducing the TiNS distance at the local minimum of the potential. **c** Time-dependent optical images of a magnetically oriented TiNS dispersion ([TiNS] = 0.5 wt%) in a quartz cuvette (40 × 10 × 1 mm) at 25 °C in air (0.04% $CO_2$) after turning off the magnetic field.

dispersion, generating an ion gradient that progressively attenuates the electrostatic repulsion between TiNSs[36,37]. Consequently, the van der Waals attraction between TiNSs prevails, thereby reducing the TiNS distance (Fig. 1b and Supplementary Fig. 3). Indeed, during the wave propagation, the TiNS distance gradually contracted from 420 to 350 nm in 20 h, as confirmed by a microscopic reflection spectral trace (Supplementary Fig. 7). The reason why such contraction results in forming the wave can be theoretically explained by considering the deformation of an elastic lamellar structure, which is known as the Helfrich–Hurault instability (Supplementary

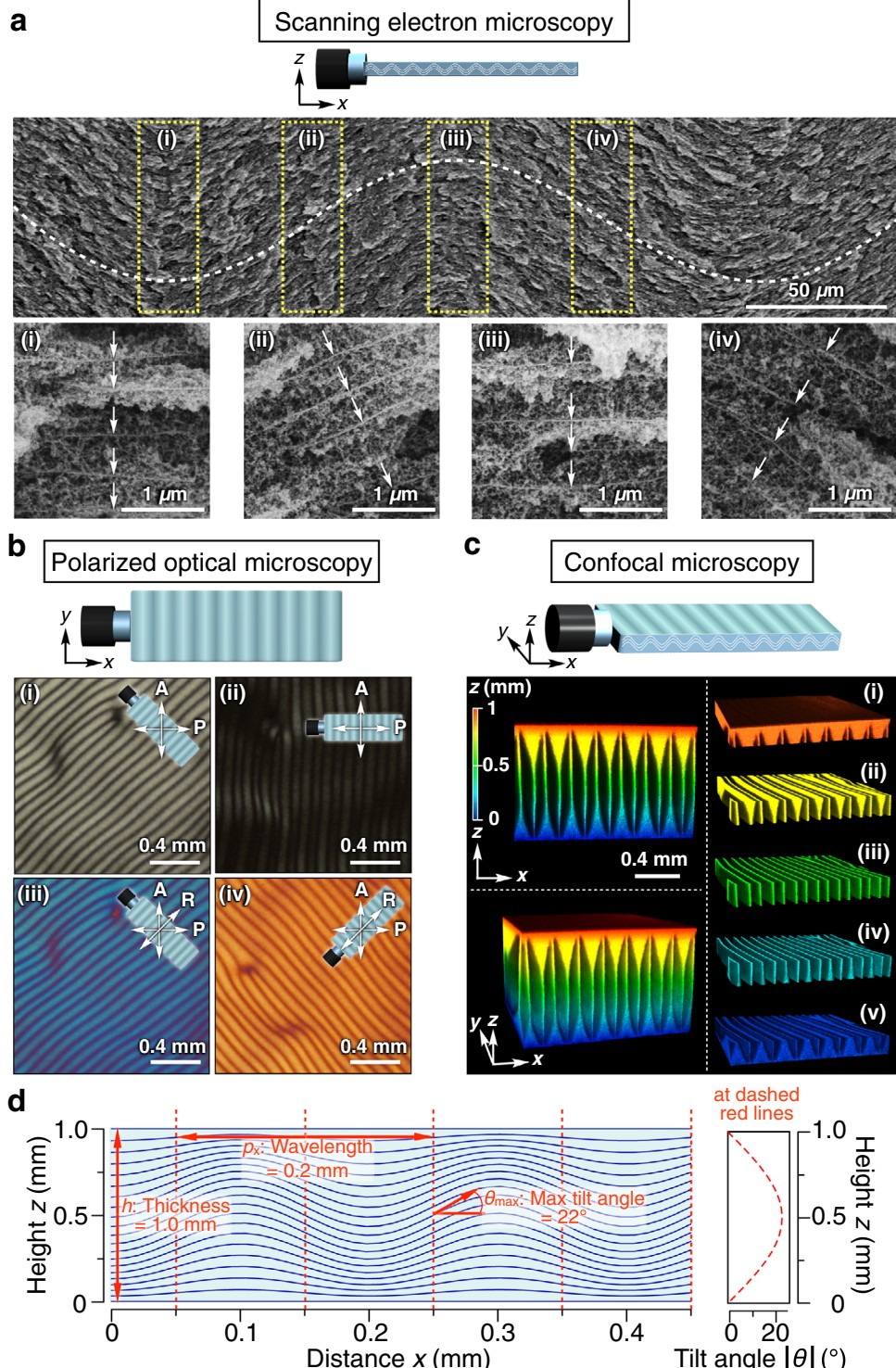

**Fig. 2 Structural characteristics of the propagating wave. a** Scanning electron microscopy (SEM) visualization of the wave structure of TiNSs, which was fixed by two-stage in situ polymerization of an acrylic monomer and a silica source, then dried, and subjected to longitudinal–sectional SEM analysis. Low-magnification (upper) and high-magnification (lower, i–iv) images. TiNSs are pointed by white arrows, while a dashed white curve represents their orientation. **b** Polarized optical microscopic (POM) images under crossed Nicols of the wave structure without (i and ii) and with (iii and iv) a sensitive tint plate. **c** Confocal laser scanning microscopic (CLSM) images in the reflection mode (488 nm laser from the top) of the wave structure. 2D images in the reflection mode were taken in a direction parallel to the *xy*-plane with a *z*-step size of 2 μm to reconstitute 3D images, where the bright regions indicate TiNSs that are oriented parallel to the *xy*-plane. 3D reconstructed image (lower left), its longitudinal-sectional view (upper left), and sliced layers at various heights (right, i–v). In **a**–**c**, a magnetically oriented TiNS dispersion of ([TiNS] = 0.5 wt%) in a quartz cuvette (40 × 10 × 1 mm) was left at 25 °C in air (0.04% $CO_2$) to generate the propagating wave. **d** Theoretically calculated wave structure with TiNSs. Left: Orientation of TiNSs in the longitudinal section of the wave structure, where the TiNS planes are directed along the sinusoidal curves. For better visibility, amplitudes of the sinusoidal curves are magnified five times. Right: Tilting angle θ of TiNSs (right) along the dashed red lines.

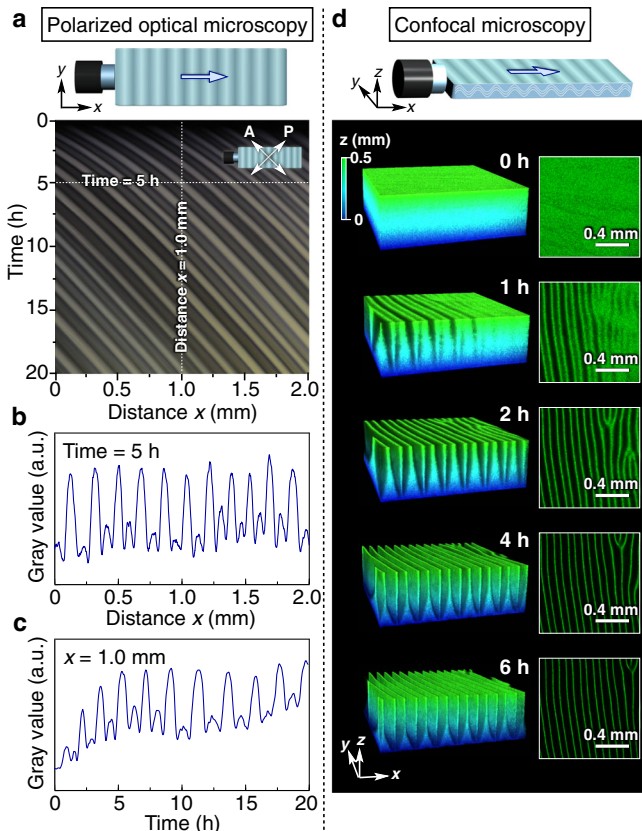

**Fig. 3 Spatiotemporal characteristics of the propagating wave. a–c** Spatiotemporal POM profile under crossed Nicols of the propagating wave (**a**) and its gray-value profiles depending on distance $x$ (**b**; at a fixed time of 5 h) and on time (**c**; at a fixed distance $x$ of 1.0 mm). **d** Time-dependent CLSM images of the propagating wave. 2D images in a reflection mode were taken in a direction parallel to the $xy$-plane with a $z$-step size of 2 µm to reconstitute 3D images, where the bright regions indicate TiNSs that are oriented parallel to the $xy$-plane. 3D reconstructed images sectioned at $z = 0.5$ mm (left) and 2D cross-sectional images at $z = 0.5$ mm (right). In **a–d**, a magnetically oriented TiNS dispersion ([TiNS] = 0.5 wt%) in a quartz cuvette ($40 \times 10 \times 1$ mm) was left at 25 °C in air (0.04% $CO_2$) to generate the propagating wave.

Method 6)[27–33]. This theoretical calculation indicates that the TiNS planes align along the sinusoidal curves in Fig. 2d. The POM image of this calculated wave, simulated using the Jones matrix method (Supplementary Fig. 8a)[40], coincides well with the experimentally observed one (Supplementary Fig. 8b), thereby supporting the wave formation mechanism described above. The Helfrich–Hurault instability[27–33] and other types of instabilities[41] were sometimes observed in liquid crystalline systems of mesogenic molecules. Such instabilities are usually induced by physical stimuli, such as electric and magnetic fields, temperature, light, and mechanical forces, whereas the present propagating wave was induced by a chemical stimulus. The key requirements for generating a macroscopic wave are that (1) the cofacial TiNS assembly must adopt a monodomain with a single crystal-like structural integrity and (2) constituent TiNSs must be strongly correlated with each other. In fact, no wave was detected without prior magnetic orientation of TiNSs into the monodomain geometry (Supplementary Fig. 9a). The same held true when the correlation of TiNSs was weakened by dilution (Supplementary Fig. 9b) or solvent exchange (ethylene glycol/water [75:25, v/ v]; Supplementary Fig. 9c). The threshold contraction of the TiNS distance for wave formation was determined to 3.7% by microscopic spectroscopy measurements (Supplementary Fig. 10), while the

bending elastic modulus $K$ was estimated to be ~1.1 pN by dynamic light scattering (DLS; Supplementary Fig. 11)[27].

**Tunability of wavelength and velocity of the propagating wave.** The wavelength and velocity of the propagating wave could be finely tuned by keeping the defect density low. As suggested by the above theoretical calculation for the wave formation, we confirmed that the wavelength was proportional to the square root of the cuvette thickness. For example, when the wave was generated in cuvettes with different thicknesses, ranging from 0.2 to 3.5 mm, its wavelength varied from 133 to 411 µm (Fig. 4a), where the characteristic penetration length (bending elastic modulus/compressive elastic modulus)$^{0.5}$ was calculated as ~3.8 µm (Fig. 4b). Meanwhile, the velocity of the propagating wave should be correlated with the diffusion velocity of ions that can be tuned by the concentration gradient of ions or the viscosity of medium. When the wave was generated by using aqueous HCl solutions with different concentrations, ranging from 0.05 to 7.5 mM, its velocity varied from 151 to 960 µm h$^{-1}$ in accordance with the variation in ion concentration gradient (Fig. 4c, d and Supplementary Fig. 12a). Indeed, the time scale of wave propagation was almost identical to that of ion diffusion, as visualized by a pH indicator dye (Supplementary Fig. 12b). The wave velocity was reduced when the viscosity of the dispersion was increased by using aqueous dispersions at different TiNS concentrations, ranging from 0.5 to 0.8 wt% (Supplementary Fig. 13).

**Directional mass transport by the propagating wave.** One of the most elegant waves in nature is the metachronal wave of cilia, which propagates in fluidic media to precisely transport mass over a long distance in a well-controlled manner[42,43]. We conceived that our propagating wave, generated by the coherent motion of ~2.4 × 10$^{10}$ individual pieces of colloidally dispersed TiNSs, would have mechanical energy to transport mass. To confirm this, we added fluorescently labeled polymer microparticles (diameter = 10 µm) to the monodomain structure of TiNSs ([TiNS] = 0.5 wt%; cuvette thickness = 1.0 mm) and conducted combined reflection/fluorescence CLSM monitoring. As shown in Fig. 5 and Supplementary Movie 6, all the microparticles were successfully transported by the propagating wave. Importantly, this transportation occurred unidirectionally (Fig. 5c) at a uniform velocity (Fig. 5d). The velocity of transportation was identical to that of the propagating wave, irrespective of whether the microparticles were modified with $NH_2$ and $CO_2H$ groups or not (Supplementary Figs. 14 and 15) and whether their diameters were 5, 10, or 20 µm (Supplementary Figs. 16 and 17). When the cofacial assembly of TiNSs was not a monodomain, neither wave generation nor transportation of microparticles was detected (Supplementary Fig. 18).

In summary, by strongly correlating the motion of billions of nanosheets dispersed in a fluidic medium, we succeeded in realizing an autonomously propagating wave that can transport mass in one direction. It is intriguing that the nanosheets, which are forced to adopt a monodomain cofacial assembly, are far apart (420 nm) from each other but strongly correlated due to competitive electrostatic repulsion and van der Waals attraction. We are sure that our finding may provide a general principle for the design of macroscopically movable devices from huge numbers of tiny components[20–22].

## Methods

**General and materials**. A Hitachi model CF16RXII centrifuge with a Hitachi model T15A41 rotor was used for the deionization of TiNS dispersions. A JASTEC model JMTD-10T100 superconducting magnet with a vertical bore of 100 mm was used for magnetic orientation of TiNSs. Optical microscopy was performed on a KEYENCE model VHX-5000 digital microscope. SEM was performed on a Hitachi model SU8010 field-emission SEM. POM was performed on a Nikon model Eclipse

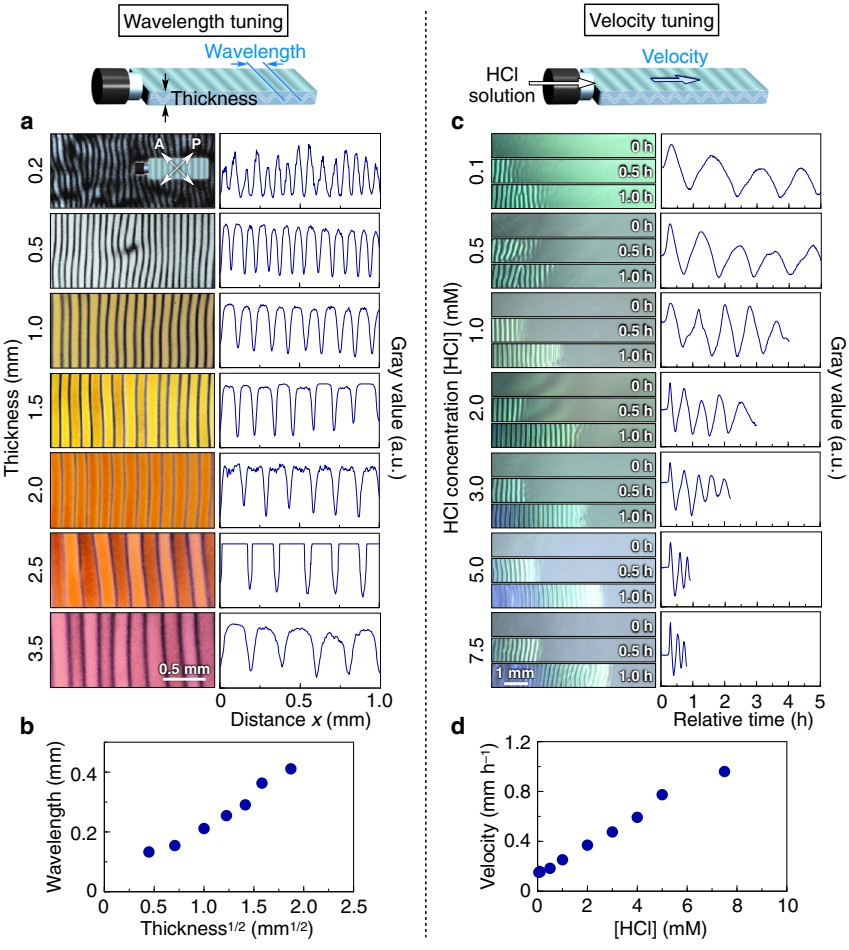

**Fig. 4 Tunability of wavelength and velocity of the propagating wave. a, b** Tuning of the wavelength by the thickness of the cuvette. Magnetically oriented TiNS dispersions ([TiNS] = 0.5 wt%) in quartz cuvettes (40 × 10 mm) with different thicknesses (0.2–3.5 mm) were left at 25 °C in air (0.04% $CO_2$) and monitored by POM under crossed Nicols. Time-dependent POM images (left) and gray-value profiles of their red channel images (right) depending on distance x (**a**) and plot of the wavelength of the propagating wave as a function of the square root of the cuvette thickness (**b**). **c, d** Tuning of the propagation velocity by the concentration of an externally added HCl solution. Magnetically oriented TiNS dispersions ([TiNS] = 0.5 wt%) in quartz cuvettes (40 × 10 × 1 mm) were treated with aqueous HCl solutions with different concentrations (0.1–7.5 mM) and monitored at 25 °C. Time-dependent optical images (left) and their gray-value profiles (right) at a fixed position (**c**) and plot of the velocity of the propagating wave as a function of the concentration of the added HCl solution (**d**).

LV100POL optical polarizing microscope. CLSM was performed on a Leica TCS SP8 confocal microscope system. Reflection spectra were recorded on a microscopic visible/near infrared (Vis/NIR) spectrometer composed of a Nikon model Eclipse LV100POL optical polarizing microscope equipped with an Ocean Optics model USB4000 spectrometer. For retardation measurement, an Olympus model U-CBE Berek compensator and an Olympus 43IF550W45 interference green filter (550 nm) was used. For DLS measurement, a Hamamatsu model H7421-40 photon counting head was used for detecting the scattered light at various scattering angles and an LS Instruments model LSI correlator for processing the signal pulses to obtain autocorrelation functions. For analysis of POM and CLSM images, the software ImageJ (http://imagej.nih.gov/ij/) was used. An aqueous dispersions of unilamellar titanate(IV) nanosheets (TiNSs; Supplementary Fig. 1) was prepared according to the literature method[35]. Water was obtained from a Millipore model Milli-Q integral water purification system. Fluorescently labeled polymer microparticles (micromer®-redF; diameter = 5, 10, and 20 μm; $CO_2H$ and $NH_2$ modified and unmodified) were purchased (Micromod Partikeltechnologie GmbH) and used after deionization.

**Deionization of an aqueous dispersion of TiNSs**. Typically, an aqueous dispersion (40 mL) of TiNSs ([TiNS] = 0.4 wt%) was centrifuged at 20,000 × g at 25 °C for 1 h. The supernatant (~35 mL) was removed from the centrifugation tube, and the residue was redispersed in the same volume of water. This procedure was repeated ~10 times to afford an aqueous dispersion of deionized TiNSs. After this deionizing procedure, the electrostatic repulsive force between TiNSs was drastically enhanced, resulting in an increase in the TiNS distance enough to exhibit a vivid structural color[37].

**Magnetic orientation of TiNSs in an aqueous dispersion**. Typically, a quartz cuvette (40 × 10 × 1 mm) filled with an aqueous dispersion of TiNSs ([TiNS] = 0.5 wt%, ~1 mL) was placed in the bore of a superconducting magnet (10 T) at 70 °C for 30 min such that the 1-mm side of the cuvette was directed parallel to the magnetic flux lines. The cuvette was then allowed to be cooled down to 25 °C for ~1 h, affording a single crystal-like monodomain cofacial assembly of colloidally dispersed TiNSs that are oriented parallel to the cuvette surface[37].

**Theoretical calculation of the TiNS distance in an aqueous dispersion**. According to the DLVO theory, the total potential ($V_{total}$) of a pair of colloidal particles is given by the sum of the potential energy of a van der Waals attractive force ($V_A$) and that of the electrostatically repulsive force ($V_R$) between the colloidal particles[36,37]:

$$V_{total} = V_A + V_R$$

For cofacially oriented TiNSs, $V_A$ and $V_R$ are expressed as follows,

$$V_A = -\frac{A}{12\pi}\left\{\frac{1}{d^2} + \frac{1}{(d+2\delta)^2} - \frac{2}{(d+\delta)^2}\right\}$$

$$V_R = \frac{64 N_A I k_B T}{\kappa}\left\{\tanh\left(\frac{e\psi_0}{4 k_B T}\right)\right\}^2 \exp(-\kappa d)$$

$\delta$: thickness of the nanosheet (=$0.75 \times 10^{-9}$ m)
$A$: Hamaker constant (=$1.0 \times 10^{-19}$ J)
$N_A$: Avogadro constant (=$6.02 \times 10^{23}$ mol$^{-1}$)
$k_B$: Boltzmann constant (=$1.38 \times 10^{-23}$ J K$^{-1}$)

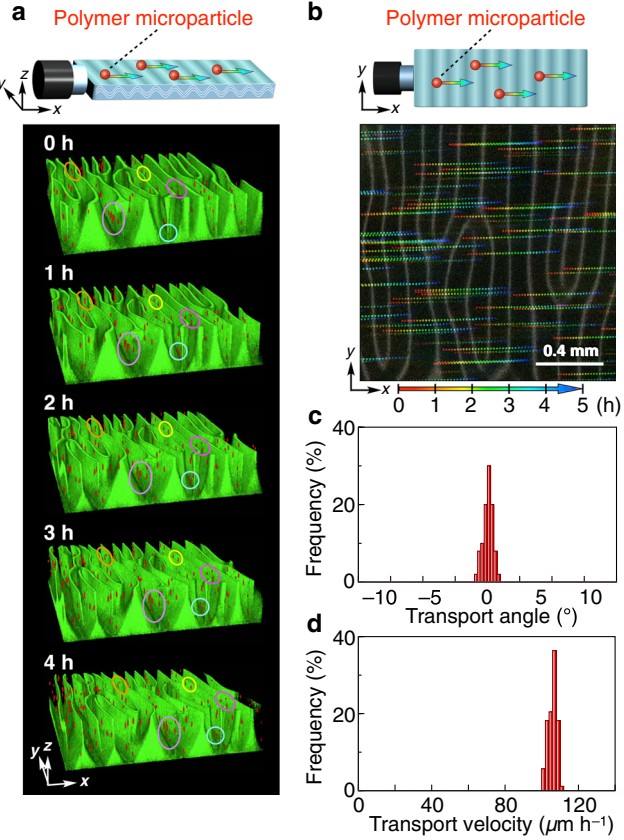

**a** Polymer microparticle

**b** Polymer microparticle

**c**
Frequency (%) — Transport angle (°)

**d**
Frequency (%) — Transport velocity ($\mu m\ h^{-1}$)

**Fig. 5 Directional mass transport by the propagating wave. a** Time-dependent 3D reconstructed CLSM images sectioned at $z = 0.5$ mm of a propagating wave (green) that transported polymer microparticles (red). A magnetically oriented TiNS dispersion ([TiNS] = 0.5 wt%; ~5 μm in width, 0.75 nm in thickness) containing fluorescently labeled polymer microparticles (10 μm in diameter; $CO_2H$ modified) in a quartz cuvette (40 × 10 × 1 mm) was left at 25 °C in air (0.04% $CO_2$). 2D images were taken in a direction parallel to the $xy$-plane with a $z$-step size of 2 μm to reconstruct 3D images, where TiNSs and the microparticles were visualized by reflection (488 nm laser) and fluorescence (522 nm laser), respectively. **b** Typical trajectories of the microparticles at 10-min intervals obtained from time-dependent 2D cross-sectional CLSM images at $z = 0.5$ mm. **c, d** Histograms of the angle (**c**) and velocity (**d**) of microparticle transport, where the angle is given relative to the wave propagating direction.

$T$: temperature (=298 K)
$e$: charge of an electron (=$1.60 \times 10^{-19}$ C)
$\psi_0$: surface potential (=−75 mV)
$I$: free-ion concentration
where the Debye screening length ($\kappa^{-1}$) is given as follows:

$$\kappa^{-1} = \sqrt{\frac{\varepsilon_r \varepsilon_0 k_B T}{2 N_A e^2 I}}$$

$\varepsilon_r$: relative permittivity of water (=80)
$\varepsilon_0$: permittivity of vacuum (=$8.85 \times 10^{-12}$ C $V^{-1}$ $m^{-1}$)

In principle, colloidal particles are positioned at the secondary local minimum of the potential curve $V_{total}$, which determines the TiNS distance. In the present case, the potential energy of the van der Waals attractive force ($V_A$) is almost unchanged while that of the electrostatically repulsive force ($V_R$) can be easily changed depending on conditions such as the free-ion concentration. Exposure of the TiNS dispersion to $CO_2$ increases the free-ion concentration of the TiNS dispersion. When the free-ion concentration of the TiNS dispersion increases from 0.27 to 0.32 mM, the calculated TiNS distance using the above equations decreases from 422 to 382 nm (Fig. 1b and Supplementary Fig. 3).

**Estimation of the TiNS distance from the reflection spectrum.** A TiNS dispersion, deionized and magnetically treated as descried above, contains TiNSs that adopt a monodomain structure with a uniform distance between the nanosheets and therefore reflects light with a specific wavelength[37]. According to the Bragg's law, the TiNS distance can be estimated from the refraction spectrum of the TiNS dispersion. Therefore, to estimate time-dependent changes of the TiNS distance, the reflection spectrum of an aqueous dispersion of TiNS ([TiNS] = 0.5 wt%) filled in a quartz cuvette (40 × 10 × 1 mm) treated with a 10-T magnetic field was measured at a selected area of the dispersion, using a microscopic Vis/NIR spectrometer (Supplementary Figs. 7b and 10b). By using the first- or second-order peaks ($m = 1$ or 2) in the reflection spectra, together with the following equation and parameters, the TiNS distances were calculated (Supplementary Figs. 7c and 10c).

$$m\lambda = 2 n_{av} d \sin\theta$$

$m$: the order of reflection (=1, 2, 3…)
$\lambda$: reflection wavelength
$\theta$: angle between the incident light and the TiNS planes (=90°)
$d$: TiNS distance
$n_{av}$: averaged reflective index of the TiNS dispersion (=1.3)

**Theoretical calculation of deformation of an elastic lamellar structure.** The reason why the contraction of the TiNS distance results in forming the wave can be theoretically explained by considering the deformation of an elastic lamellar structure[27–33]. For the initial state of our theoretical model, we consider an elastic lamellar structure confined between two parallel flat surfaces separated by a distance $h$, with the lamellar layers oriented parallel to the $xy$-plane. When an external stimulus is applied to the lamellar structure, a local displacement of the layers, denoted by $u(x, z)$, is induced. The free energy density $F_d$ of the system in a volume of $V$ can be expressed as:

$$F_d = \int_V \frac{1}{2}\left[B\left(\frac{\partial u}{\partial z} - \frac{\theta^2}{2}\right)^2 + K\left(\frac{\partial^2 u}{\partial x^2}\right)^2\right]dV.$$

$V$: volume of the lamellar layers
$\theta$: tilting angle of the lamellar layers (= $\partial u/\partial x$)
$B$: compressive elastic modulus of the lamellar layers
$K$: bending elastic modulus of the lamellar layers

Minimization of the free energy density $F_d$ gives a solution for the layer displacement, $u(x, z)$, as follows, where $u_0$ represents amplitude:

$$u = u_0 \sin\left(\frac{\pi}{h}z\right)\cos\left(\frac{\pi}{\left[\pi(K/B)^{0.5}h\right]^{0.5}}x\right)$$

In order to correlate this solution with the wave structure with TiNSs, we introduce characteristic parameters $p_x\left(=2\left[\pi(K/B)^{0.5}h\right]^{0.5}\right)$ and $\theta_{max}$ $\left(=-\pi u_0/\left[\pi(K/B)^{0.5}h\right]^{0.5}\right)$, which correspond to the wavelength of the propagating wave and the maximum tilting angle of TiNSs, respectively. Thus, $\theta = \partial u/\partial x$ can be expressed as follows:

$$\theta = \frac{\partial u}{\partial x} = \theta_{max}\sin\left(\frac{\pi}{h}z\right)\sin\left(\frac{2\pi}{p_x}x\right)$$

In the present case ([TiNS] = 0.5 wt%; $h = 1$ mm; $p_x = 200$ μm; $\theta_{max} = 22°$), this equation represents that the TiNS planes align along the sinusoidal curves in Fig. 2d. As described above, the wavelength of the propagating wave follows the equation $p_x = 2\left[\pi(K/B)^{0.5}h\right]^{0.5}$. Indeed, the wavelength is proportional to the square root of the cuvette thickness (Fig. 4b), where the characteristic penetration length, $(K/B)^{0.5}$, was calculated as ~3.8 μm.

## Data availability
The authors declare that the data supporting the findings of this study are available within the article and its supplementary information files. All other information is available from the corresponding authors upon reasonable request.

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

## Acknowledgements

This work was supported by JST PRESTO Grant Number JPMJPR20A6, Japan (to K.S.), JST CREST Grant Number JPMJCR17N1, Japan (to Y.I.), and JSPS KAKENHI Grant Numbers JP19K23642 (to K.S.), JP20K15350 (to K.S.), JP18H05260 (to T.A.). K.S. acknowledges the Kurita Water and Environment Foundation (KWEF, Japan) and the RIKEN Special Post-doctoral Researcher Program. We thank Mr. Kiyoshi Morishita for English proofreading.

## Author contributions

K.S. designed and performed all the experiments. Y.I. and T.A. co-designed the experiments. X.W. and Z.S. supported electron microscopic observations. S.A. and F.A. performed dynamic light scattering set-up and analysis, theoretical calculations, and simulations. Y.E. and T.S. prepared colloidally dispersed TiNSs. K.S., S.A., F.A., Y.I. and T.A. analyzed the data. K.S., Y.I. and T.A. wrote the manuscript, and all authors contributed to the final manuscript.

## Competing interests

The authors declare no competing interests.
