## [Peer Review File · Nature Communications]

Propagating wave in a fluid by coherent motion of 2D colloidsEditorial Note: This manuscript has been previously reviewed at another journal that is not operating a transparent peer review scheme. This document only contains reviewer comments and rebuttal letters for versions considered at *Nature Communications*.

REVIEWER COMMENTS

Reviewer #2 (Remarks to the Author):

This is an interesting work showing how propagating wave motion can be generated in a pre-aligned colloidal liquid-crystal-like material of dispersed charge-repulsive nanosheets in aqueous solution. The authors have performed several different systematic studies in order to identify the nature of the propagating instability, and these include control experiments to show a clear link to ionic content in the aqueous phase. The movies of the effect, linked to a Helfrich-Hurault (H-H) instability, are particularly striking and beautiful. Overall, I think that the authors' submission is suitable for publication in *Nature Communications* after some significant modifications have been made to the main text to help the reader better understand the origins and limits of the effect that the authors have explored.

First, the use of the term 'interlocking' is confusing and even a bit misleading; so, readers would benefit if better terminology would be used by the authors instead. The nanosheets are charge-repulsive colloids, and their main interaction is via screened-charge repulsion at high density, not interlocking. For example, if more water is added to the authors' system, the repulsion will cause the charged nanosheet particles to move apart from each other and dilate to fill the new system volume uniformly, provided that the Debye screening length is sufficiently large. Such behavior is not characteristic of interlocking; a truly interlocked system would not exhibit such dilation upon dilution. So, only certain physical systems that are truly interlocking deserve this designation, and the authors' system does not appear to meet this requirement. For instance, charge-stabilized aqueous dispersions of nanospheres at high densities and low ionic strengths also don't interlock but do exhibit interesting behavior, too, particularly when the screening length is comparable to or exceeds the average distance between surfaces of the nanospheres. So, the authors' system would be better categorized as a dispersed charge-repulsive colloidal plate system reminiscent of a discotic molecular liquid crystal (LC), not an interlocking system. The authors explain their observed propagating instability using H-H notions, which is reasonable; so, why not further link their system to what is already known for molecular discotic LCs? Their pre-aligned colloidal phase is smectic type, if the analogy to H-H holds. What is the bending elasticity of the smectic based on H-H theory as a function of ionic strength? This could potentially be deduced or estimated from their optical imaging.

The authors' work needs to be considered in light of prior work decades ago by Pleiner, Stannarius, and Zimmermann on driven instabilities in smectic films ("Evolution of Structures in Continuous Dissipative Systems", eds. F. Busse and S.C. Müller, Lecture Notes in Physics, Springer Berlin, p.295 (1998)). While these studies involved electrical driving, many of the patterns formed are similar enough to be noted, and dynamic aspects of the instabilities were considered in thin films, analogous to what the authors have done in a cell. The authors need to cite this work and comment on different forms of drive for instability motion (e.g. ionic versus electric).

Although systematic studies have been conducted, which strengthens the authors' findings overall, there are several limits on these studies that are not sufficiently explained. First, regarding cell thickness, why not go to 50 micron, 100 micron, 250 micron, and 500 micron thick cells, since these are readily available? Having four points on a plot (Fig 4b) is better than not conducting the study at all, but the lower and upper limits seem to be chosen over a very small dynamic range, when other options would be readily available (both smaller thickness and larger thickness). The authors have not done a large enough of a dynamic range in thickness to claim a power law of $1/2$, even if the data taken so far might be consistent with that. Also, again for [HCl], why limit this to such a small range? Is there a threshold effect in [HCl] required to induce the instability? Or, is the instability found, even if to lesser and lesser rotation amplitudes, all the way down to minute [HCl]?

Presumably, the elastic constant associated with the rotation amplitude of the plates for the H-H instability depends on the volume fraction of nanosheets and on the ionic concentration, which varies locally in the authors' system. From their data, can the authors make a clear plot of this elasticity as a function of screening length relative to average spacing between nanosheets,

derived from all data? Readers would find this to be useful.

It seems that the instability results from the gradient in ion concentration, not the ion concentration itself. Gradients in osmotic pressure are known to drive flows (e.g. of microparticles) in colloidal systems already, independent of instabilities related to repulsive nanosheet orientations. There is no need to refer to machines if gradient-driven flow is really the underlying causation and the cool-looking propagating H-H instability is just a co-effect. So, to really claim something like a machine, the authors are really stretching quite a bit. Many other physical mechanisms can more efficiently move microparticles at much rates greater than 0.04 microns per second (equivalent of 150 microns/hour). It just seems like the introduction of the paper could be re-written with less hype about machines and more focus on what is already known about propagating H-H instabilities known from driven molecular LCs.

Clear comparison of the Debye screening length (or other useful measure related to charge repulsion) relative to spacing between platelets (normal to their faces) would be instructive, rather than referring to ionic concentrations directly. While the visualizations capture much useful information, it would be useful to know the instantaneous ionic concentrations as a function of space and time too. At least for the HCl experiments, it might be possible to put a pH-sensitive dye of some kind into the aqueous phase that could be used to measure this directly.

There is a typo on p. 4: change 'defused' to 'diffused'

In summary, the authors' experiments are well-conceived overall, and these beautifully demonstrate a dynamic H-H instability in a colloidal repulsive platelet system. If the authors suitably modify their presentation and discussion in line with the above comments, particularly focusing on the main text, then a suitably revised manuscript would warrant publication in Nature Communications.

Answers to Comments Raised by Reviewer

For Reviewer #2

This is an interesting work showing how propagating wave motion can be generated in a pre-aligned colloidal liquid-crystal-like material of dispersed charge-repulsive nanosheets in aqueous solution. The authors have performed several different systematic studies in order to identify the nature of the propagating instability, and these include control experiments to show a clear link to ionic content in the aqueous phase. The movies of the effect, linked to a Helfrich-Hurault (H-H) instability, are particularly striking and beautiful. Overall, I think that the authors' submission is suitable for publication in Nature Communications after some significant modifications have been made to the main text to help the reader better understand the origins and limits of the effect that the authors have explored.

=> We appreciate these highly encouraging comments.

(1) First, the use of the term 'interlocking' is confusing and even a bit misleading; so, readers would benefit if better terminology would be used by the authors instead. The nanosheets are charge-repulsive colloids, and their main interaction is via screened-charge repulsion at high density, not interlocking. For example, if more water is added to the authors' system, the repulsion will cause the charged nanosheet particles to move apart from each other and dilate to fill the new system volume uniformly, provided that the Debye screening length is sufficiently large. Such behavior is not characteristic of interlocking; a truly interlocked system would not exhibit such dilation upon dilution. So, only certain physical systems that are truly interlocking deserve this designation, and the authors' system does not appear to meet this requirement. For instance, charge-stabilized aqueous dispersions of nanospheres at high densities and low ionic strengths also don't interlock but do exhibit interesting behavior, too, particularly when the screening length is comparable to or exceeds the average distance between surfaces of the nanospheres. So, the authors' system would be better categorized as a dispersed charge-repulsive colloidal plate system reminiscent of a discotic molecular liquid crystal (LC), not an interlocking system.

=> Thank you for this constructive suggestion. In response to this suggestion, we removed the term 'interlocking' and revised the corresponding parts of the manuscript.

(2) The authors explain their observed propagating instability using H-H notions, which is reasonable; so, why not further link their system to what is already known for molecular discotic LCs?

=> According to this comment, we added a description about the H-H instability of molecular LCs to the main text on page 3, line 8 and on page 6, line 23, citing a new reference about molecular discotic LCs (Ref. 33).

(3) Their pre-aligned colloidal phase is smectic type, if the analogy to H-H holds. What is the bending elasticity of the smectic based on H-H theory as a function of ionic strength? This could potentially be deduced or estimated from their optical imaging.

=> As Reviewer #2 suggested, we can estimate the characteristic penetration length [= (bending elastic modulus/compressive elastic modulus)^{0.5}] of the propagating wave from its wavelength observed in optical images. Meanwhile, in the propagating wave, the ionic strength gradually decreases from the ion-supplied end to the wave-front end. We can estimate the ionic strength at a certain region from the reflection spectroscopy peak at this region, based on Bragg's law and DLVO theory.

=> Accordingly, we calculated the ionic strengths and wavelengths at four regions in the propagating wave (Supplementary Fig. 10), where the ionic strengths changed from 0.31 to 0.41 mM, while the characteristic penetration lengths hardly changed (see the following table). Thus, within the ionic-strength range of 0.31–0.41 mM, which is a typical condition for causing the propagating wave, the bending elastic modulus is likely to remain almost constant. We added a related description to the caption of Supplementary Fig. 10.

Relative x position (mm)	2nd order peak (nm)	[Free ion] (mM)	Wavelength (μm)	Penetration length (μm)
0	508	0.31	239	4.5
-2	497	0.32	237	4.5
-4	459	0.37	244	4.7
-6	430	0.41	238	4.5

(4) The authors' work needs to be considered in light of prior work decades ago by Pleiner, Stannarius, and Zimmermann on driven instabilities in smectic films (“Evolution of Structures in Continuous Dissipative Systems”, eds. F. Busse and S.C. Müller, Lecture Notes in Physics, Springer Berlin, p.295 (1998). While these studies involved electrical driving, many of the patterns formed are similar enough to be noted, and dynamic aspects of the instabilities were considered in thin films, analogous to what the authors have done in a cell. The authors need to cite this work and comment on different forms of drive for instability motion (e.g. ionic versus electric).

=> Thank you for this useful suggestion. Accordingly, we newly cited the suggested reference in the main text (Ref. 41) and added a related description to the main text on page 6, line 23.

(5) Although systematic studies have been conducted, which strengthens the authors' findings overall, there are several limits on these studies that are not sufficiently explained. First, regarding cell thickness, why not go to 50 micron, 100 micron, 250 micron, and 500 micron thick cells, since these are readily available? Having four points on a plot (Fig 4b) is better than not conducting the study at all, but the lower and upper limits seem to be chosen over a

very small dynamic range, when other options would be readily available (both smaller thickness and larger thickness). The authors have not done a large enough of a dynamic range in thickness to claim a power law of $1/2$, even if the data taken so far might be consistent with that.

=> In response to this comment, we expanded the cell-thickness range (0.2–3.5 mm; Fig. 4a) and confirmed that the wavelength is proportional to (cell thickness)^{1/2} within this range (Fig. 4b). Accordingly, we revised the corresponding parts of the manuscript.

(6) Also, again for [HCl], why limit this to such a small range? Is there a threshold effect in [HCl] required to induce the instability? Or, is the instability found, even if to lesser and lesser rotation amplitudes, all the way down to minute [HCl]?

=> In response to this comment, we expanded the [HCl] range (0.05–10 mM; Fig. 4c, d and Supplementary Fig. 12a). Accordingly, we revised the corresponding parts of the manuscript.

(7) Presumably, the elastic constant associated with the rotation amplitude of the plates for the H-H instability depends on the volume fraction of nanosheets and on the ionic concentration, which varies locally in the authors' system. From their data, can the authors make a clear plot of this elasticity as a function of screening length relative to average spacing between nanosheets, derived from all data? Readers would find this to be useful.

=> Within the ionic-strength range of 0.31–0.41 mM, which is a typical condition for causing the propagating wave, the bending elastic modulus is likely to remain almost constant, as described in our answer to comment (3). For the relationship between the free-ion concentration, Debye screening length, and TiNS distance, please see our answer to comment (9).

(8) It seems that the instability results from the gradient in ion concentration, not the ion concentration itself. Gradients in osmotic pressure are known to drive flows (e.g. of microparticles) in colloidal systems already, independent of instabilities related to repulsive nanosheet orientations. There is no need to refer to machines if gradient-driven flow is really the underlying causation and the cool-looking propagating H-H instability is just a co-effect. So, to really claim something like a machine, the authors are really stretching quite a bit. Many other physical mechanisms can more efficiently move microparticles at much rates greater than 0.04 microns per second (equivalent of 150 microns/hour). It just seems like the introduction of the paper could be re-written with less hype about machines and more focus on what is already known about propagating H-H instabilities known from driven molecular LCs.

=> On the basis of these suggestions, we removed the term 'machine' and revised the corresponding parts of the manuscript.

(9) Clear comparison of the Debye screening length (or other useful measure related to charge repulsion) relative to spacing between platelets (normal to their faces) would be instructive, rather than referring to ionic concentrations directly.

=> In response to this suggestion, we described the relationship between the free-ion concentration, Debye screening length, and TiNS distance (Supplementary Fig. 3).

(10) While the visualizations capture much useful information, it would be useful to know the instantaneous ionic concentrations as a function of space and time too. At least for the HCl experiments, it might be possible to put a pH-sensitive dye of some kind into the aqueous phase that could be used to measure this directly.

=> According to this helpful suggestion, we used a bromothymol blue (BTB) solution to trace the pH change after the introduction of the HCl solution ($[HCl] = 1 \text{ mM}$). We confirmed that the BTB solution gradually changed its color upon the HCl diffusion, which was in good agreement with the wave-generation experiment (Supplementary Fig. 12b). We added a related description to the main text on page 8, line 1.

(11) There is a typo on p. 4: change 'defused' to 'diffused'

=> According to this suggestion, we changed the word 'defused' to 'diffused' in the main text on page 4, line 16.

In summary, the authors' experiments are well-conceived overall, and these beautifully demonstrate a dynamic H-H instability in a colloidal repulsive platelet system. If the authors suitably modify their presentation and discussion in line with the above comments, particularly focusing on the main text, then a suitably revised manuscript would warrant publication in Nature Communications.

=> We appreciate these encouraging comments. Owing to the additional experiments and the revision of manuscript for addressing the constructive comments and suggestions raised by Reviewer #2, we believe that our manuscript has now been considerably polished up.